# M³PC: Test-time Model Predictive Control for Pretrained Masked Trajectory Model

**Kehan Wen**[1†], **Yutong Hu**[1,2†], **Yao Mu**[3*], **Lei Ke**[4*]

[1]ETH Zurich, [2]KU Leuven, [3]Hong Kong University, [4]Carnegie Mellon University

## Abstract

Recent work in Offline Reinforcement Learning (RL) has shown that a unified Transformer trained under a masked auto-encoding objective can effectively capture the relationships between different modalities (e.g., states, actions, rewards) within given trajectory datasets. However, this information has not been fully exploited during the inference phase, where the agent needs to generate an optimal policy instead of just reconstructing masked components from unmasked ones. Given that a pretrained trajectory model can act as both a Policy Model and a World Model with appropriate mask patterns, we propose using Model Predictive Control (MPC) at test time to leverage the model's own predictive capability to guide its action selection. Empirical results on D4RL and RoboMimic show that our inference-phase MPC significantly improves the decision-making performance of a pretrained trajectory model without any additional parameter training. Furthermore, our framework can be adapted to Offline to Online (O2O) RL and Goal Reaching RL, resulting in more substantial performance gains when an additional online interaction budget is provided, and better generalization capabilities when different task targets are specified. Code is available: https://github.com/wkh923/m3pc.

## 1 Introduction

The Masked Modeling paradigm has a simple, self-supervised training objective: predicting a randomly masked subset of the original sequence. It has become a powerful technique for generation or representation learning for sequential data, e.g., language tokens (Devlin et al., 2018) or image patches (He et al., 2022). Unlike autoregressive models like GPT (Brown et al., 2020), which condition only on the past context in the "left", bidirectional models trained with this objective learn to model the context from both sides, leading to richer representations and deeper understandings of the data's underlying dependencies.

Given that a sequential decision-making trajectory inherently involves a sequence of states $s$ and actions $a$, and other optional augmented properties like return-to-go (RTG) $g$ (Chen et al., 2021) or approximate state-action value $v$ (Yamagata et al., 2023) across $T$ timesteps, the mask modeling paradigm can be adapted easily for sequential decision-making tasks. For example, in the case of Reinforcement Learning, the policy output $\mathbb{P}(a|s)$ at each time step can be regarded as predicting a masked action $a$ conditioned on given states $s$. Moreover, recent works (Carroll et al., 2022; Liu et al., 2022; Wu et al., 2023) have demonstrated that a unified bidirectional trajectory model (BTM) pretrained with a highly random masking pattern can be applied zero-shot in various downstream tasks. Different reconstruction tasks can be deliberately created by applying appropriate masks to different modalities- whether states, actions, or rewards- during inference time.

However, the inherent flexibility and versatility of models trained with random masking techniques have not been fully exploited in deployment settings. Previous research has highlighted the multitasking capabilities of Bidirectional Trajectory Models (BTMs) by assigning **one** single specific mask pattern to individual tasks, such as the RCBC mask commonly used in offline RL after the pretraining phase. Our findings, in contrast, suggest that integrating **multiple** capabilities such as short-term reward and long-term return prediction, along with forward dynamics, could significantly

---

†Equal contribution. *Corresponding authors. This work was done at ETH Zurich.

Figure 1: **Benefits of equipping pretrained bidirectional trajectory model with our test-time M³PC.** (a) Instead of generating actions solely based on history context, we leverage the full capacity of the masked pretrained model to predict future outcomes (e.g. states, rewards, returns) as a test-time self-enhanced decision making approach. Such a MPC framework can be used to achieve higher return at inference time or to reach a given goal state (in dashed square block) even unseen during offline training. (b) Forward M³PC achieves better offline learning performances, using the same model without any finetuning, and gains better O2O improvement when online finetuning is allowed after offline pretrain. (c) Backward M³PC unlocks zero shot goal reaching capability. Given a desired state, the walker agent can split its legs to a large degree without any prior experience.

enhance decision-making. These capabilities allow the agent to explicitly evaluate action candidates and determine the optimal one, rather than merely relying on implicit mappings from expected returns to policies.

Building on these insights, we introduce the **M³PC** framework: Enhancing Decision-Making by using the **M**asked **M**odel itself for test-time **M**odel **P**redictive **C**ontrol. Our framework decomposes decision-making tasks into a series of simpler steps in a typical sample-based MPC style: sampling potential actions, inferring possible future states, evaluating these actions based on predicted outcomes, and selecting the final optimal action. We then demonstrate how a pretrained model, equipped with our adaptation and ensemble of masks, can efficiently and effectively handle these subtasks. Our empirical results demonstrate that, by using M³PC for final decision-making, the same pretrained model can get substantial decision quality improvement in offline RL and goal-reaching RL, outperforming traditional single-mask models. Furthermore, M³PC supports sample-efficient online finetuning — a capability rarely seen in previous sequential modeling agents. By fully leveraging the potential of a pretrained BTM, M³PC evolves the model from a multitasking framework into an inference phase self-enhancing, and a finetuning phase self-improving generalist agent. We summarize our results in Figure 1 and highlight our contributions as:

- We present M³PC, a novel framework that utilizes mask ensembles to address complex decision-making tasks, effectively leveraging the multitasking abilities of a pretrained bidirectional trajectory model (BTM).

- We demonstrate that M³PC not only improves the test-time performance of the same pretrained BTM in offline RL by 6.0%, but also enables efficient finetuning through online interactions with environments, outperforming specialized offline-to-online (O2O) RL algorithms, such as ODT, by 26.0%.

- We show that M³PC can be adapted for goal-reaching tasks, effectively guiding agents to specified goal states—even when these states are out-of-distribution relative to the datasets used for pretraining.

## 2 RELATED WORK

**Transformers for Sequential Decision Making.** The Transformer (Vaswani et al., 2017) architecture has been extensively applied in sequential decision-making tasks such as reinforcement learning (RL) (Chen et al., 2021; Janner et al., 2021; Wang et al., 2022) and imitation learning (IL) (Reed et al., 2022; Shafiullah et al., 2022; Brohan et al., 2022; Baker et al., 2022; Jia et al., 2023). Representative work such as Decision Transformer (DT) (Chen et al., 2021) and its variants (Zheng et al., 2022; Yamagata et al., 2023) learn a return-conditioned policy using a causal-masked Transformer. Recent studies (Carroll et al., 2022; Liu et al., 2022; Wu et al., 2023) utilize a bidirectional Transformer to model trajectories, highlighting the model's versatility enhanced by the mask prediction training objective. These studies focus on the potential of trajectory Transformers to unify various decision-making tasks, typically employing a unique mask pattern tailored to each specific down-

stream task. Building upon these insights, our work proposes harnessing the functional versatility of pretrained models to enhance decision-making. More specifically, we investigate whether utilizing two or more mask patterns can lead to improved decision-making within a single downstream task.

**Offline RL with Online Finetuning.** Traditional off-policy RL algorithms often suffer from bootstrapping error accumulation (Fujimoto et al., 2019; Nair et al., 2020). To mitigate these issues, most offline RL algorithms employ regularization techniques to mitigate errors caused by out-of-distribution actions (Fujimoto et al., 2019; Nair et al., 2020; Kumar et al., 2020; Kostrikov et al., 2021; An et al., 2021; Kumar et al., 2019). However, finetuning an offline RL algorithm can be challenging due to its inherent conservatism and the offline-to-online data distribution shift (Nair et al., 2020; Yu & Zhang, 2023). Many techniques such as value calibration (Nakamoto et al., 2024), balanced replay (Lee et al., 2022) and policy expansion (Zhang et al., 2023) have been investigated to improve the online sample efficiency. In parallel, some work (Chen et al., 2021; Zheng et al., 2022) following supervised learning (SL) paradigm can naturally ensure in-distribution learning but also suffer from poor online sample efficiency (Brandfonbrener et al., 2022). Our approach adheres to the SL paradigm while incorporating DP-based module to improve online sample efficiency.

**Model-based RL.** Learning a dynamics model of the environment can be used for policy learning (Pong et al., 2018; Ha & Schmidhuber, 2018; Hafner et al., 2019) or planning (Silver et al., 2008; Walsh et al., 2010; Zhang et al., 2019; Yu et al., 2020). Recent work has explored the feasibility of MPC in online RL (Chua et al., 2018; Janner et al., 2019; Wu et al., 2022; Lowrey et al., 2018; Hatch & Boots, 2021; Hansen et al., 2022) Similar planning methods have also been tailored for offline RL using techniques such as behavior cloning regularization (Argenson & Dulac-Arnold, 2020) and trajectory pruning (Zhan et al., 2021; Wang et al., 2023). Instead of maintaining separate world and policy models, Trajectory Transformer (TT) (Janner et al., 2021) frames RL as a sequential modeling problem and performs beam search planning based on return heuristics. Our work follows a similar paradigm but leverages a bidirectional Transformer and mask autoencoding to enable a more flexible and computationally efficient planning process.

## 3 PRELIMINARY

We consider the environment as a Markov Decision Process (MDP), formally defined by the tuple $\mathcal{M} = \langle \mathcal{S}, \mathcal{A}, P, R, \gamma, \rho_0 \rangle$. In this notation, $\mathcal{S}$ represents the state space, and $\mathcal{A}$ represents the action space. The transition probability distribution, $P(s_{t+1} \mid s_t, a_t)$, defines the likelihood of transitioning from state $s_t$ to state $s_{t+1}$ given action $a_t$. The reward function, $R(s_t, a_t)$, assigns a reward for each action taken in a particular state. The discount factor, denoted by $\gamma$, quantifies the preference for immediate rewards over future rewards. The maximum episode length, also referred to as the horizon of the MDP, is denoted as $H$.

Additional notations are introduced to adapt RL to sequential modeling. We denote the training data distribution as $\mathcal{T}$, which may be dynamic when the agent interacts with the environment. A trajectory $\tau$, consisting of $T$ states, actions, RTGs and rewards is represented as $\tau = (s_1, g_1, a_1, r_1, \cdots, s_T, g_T, a_T, r_T)$. Note that some other properties can also be directly or indirectly accessed from the training data such as next-states $(s'_1, \cdots, s'_T)$, estimated values $(v_1, \cdots, v_T)$ for state-action pairs, but we do not model these modalities on the Transformer.

## 4 METHOD

This section details how we leverage a bidirectional trajectory model's versatile prediction capabilities within the M³PC framework to enhance an agent's decision-making. In a typical MPC process, the system repeatedly solves an optimization problem to identify the best sequence of actions over a finite horizon by evaluating the outcomes of these actions and then executing the action at the current timestep. The following subsections describe our approach to adapting a BTM to perform these MPC steps: First, we enable the BTM to **reconstruct actions with uncertainty**, allowing us to sample from a distribution of action proposals. Next, we demonstrate how to use different masking patterns for **forward** or **backward** prediction for MDP sequence elements. These predictions serve as references for evaluating the expected outcomes of action proposals, which we use to determine the optimal action to execute. As illustrated in Figure 3, by breaking down decision-making into these structured steps and using the BTM for versatile predictions, our M³PC framework en-

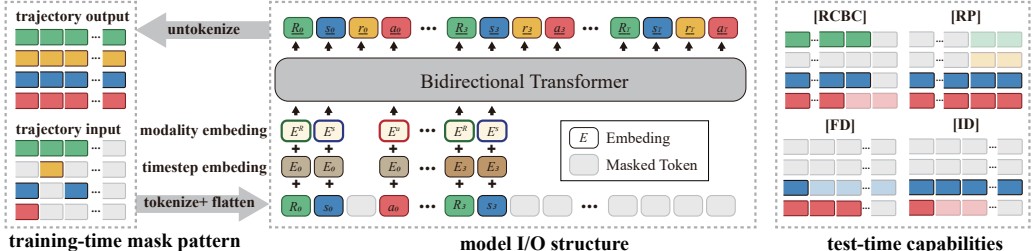

Figure 2: **Model overview.** The bidirectional trajectory model is pre-trained using MAE loss that aims to reconstruct the whole MDP trajectory taken a **[Random]** masked trajectory. After pre-training, the model show multiple capabilities by applying different test-time masks. E.g., **Return-Conditioned Behaviour Clone [RCBC] Mask:** Predict actions given states, expected return and context trajectory. **Reward and Return Prediction [RP] Mask:** Predict intermediate rewards and future return given states and actions. **Forward Dynamics [FD] Mask:** Predict future states given current state and future actions. **Inverse Dynamics [ID] Mask:** Infer actions needed taken to perform a given state path. As a pretrained masked transformer can always reconstruct the full trajectory, for those MDP-elements that are not related to the given task, e.g., the rewards during [RCBC], we omit and mark them as gray.

hances the agent's ability beyond simply imitating behaviors observed in offline data, e.g., achieving higher reward incomes or diverse goals which typically fall in offline RL and goal reaching domains, respectively.

**Bidirectional Trajectory Model.** We illustrate the model architecture and how it process a masked MDP trajectory as Figure 2. To perform masked trajectory modeling, we first flatten and tokenize the different elements of the raw trajectory sequence. This tokenization involves three components: a modality-specific encoder that lifts elements from the raw modality space to a common representation space, along with the addition of timestep embeddings and modality-type embeddings. These components collectively enable the Transformer to distinguish between different sequence elements.

We employ an encoder-decoder architecture with both the encoder and decoder being bidirectional Transformers. The tokenized and flattened trajectory is fed into the Transformer encoder, where only unmasked tokens are processed. The decoder then processes the full trajectory sequence, leveraging values from the encoder when available or substituting a mask token when not. The decoder is trained to reconstruct the original sequence, including the unmasked tokens.

**Training-phase Mask Pattern.** Inspired by previous work (Wu et al., 2023; Zeng et al., 2024), we employ a two-step masking pattern for training. Firstly, we randomly mask a proportion of elements in the trajectory $\tau$. Secondly, we mask all elements to the right of a randomly chosen position. By learning to predict the mask elements, the model is trained to handle temporal dependencies and infer outcomes based solely on past events.

**Uncertainty-Aware Action Reconstruction.** To equip the agent with robust decision-making capabilities beyond mere imitation, our method employs uncertainty-aware action reconstruction rather than predicting the masked action deterministically. The primary focus of MAE lies in perfectly reconstructing each token of the sequence, typically by optimizing a Mean Squared Error (MSE) loss. This inherently leads to deterministic action reconstruction, which limits the agent's ability to account for uncertainties associated with the actions.

To address this limitation, we propose reconstructing an uncertainty-aware action distribution $\mathcal{A}$ by minimizing a Negative Log Likelihood (NLL) loss $J(\theta)$ denoted by

$$J(\theta) = \frac{1}{T}\mathbb{E}_{\tau \sim \mathcal{T}}\left[\sum_{t=1}^{T} -\log P_\theta(a_t|\texttt{Masked}(\tau))\right]. \qquad (1)$$

Inspired by ODT (Zheng et al., 2022), we additionally impose a lower bound on trajectory-level action entropy $\mathcal{H}_\theta^{\mathcal{T}}$ to encourage the agent's online exploratory behavior. The overall constraint problem is formally defined as

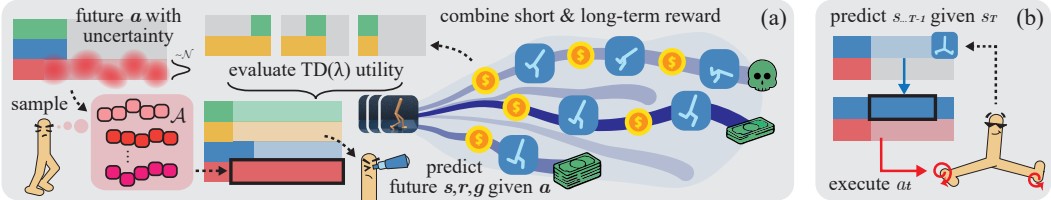

Figure 3: **Leverage the Masked Model itself for test-time Model Predictive Control.** Our pipeline utilizes BTM's versatile inference capabilities to enhance decision making. **(a) Forward M³PC.** We employ [RCBC], [FD] and [RP] masks to build an MPC pipeline for planning, prediction, and action resample. **(b) Backward M³PC.** Given a goal state that we finally want to reach, we first use Path Inference [PI] mask to infer the waypoint-states, followed by a Inverse Dynamic [ID] mask to get the action sequence conditioned on those waypoints, and finally execute the first one.

$$\min_{\theta} J(\theta) \text{ subject to } \mathcal{H}_{\theta}^{\mathcal{T}} \geq \beta, \ \mathcal{H}_{\theta}^{\mathcal{T}} = \frac{1}{T}\mathbb{E}_{\tau \sim \mathcal{T}}\left[\sum_{t=1}^{T} H\left[P_{\theta}(a_t|\texttt{Masked}(\tau))\right]\right], \quad (2)$$

where $H[\cdot]$ denotes the Shannon entropy of the distribution, $\beta$ represents the predefined target entropy. To avoid explicitly handling the inequality constraint, we solve the Lagrangian dual problem of Equation 2. Implementation details are provided in Appendix A.

**Forward M³PC for Reward Maximization.** A bidirectional trajectory model agent has demonstrated zero-shot ability in offline RL tasks when equipped with an [RCBC] mask as shown in previous work (Carroll et al., 2022; Wu et al., 2023). By predicting actions conditioned on states and RTGs, the agent generates actions by imitating trajectories with similar RTGs in offline data. The performance of this imitative behavior is inherently upper-bounded by the best trajectory in the offline data.

To address this limitation, we propose refining the decision-making process by implementing an explicit reward-maximization procedure using the forward dynamics function and the return and reward prediction functions provided by the unified trajectory model. Typically, this process divides decision-making into three substeps: generating action proposals, rolling out the future, and selecting action proposals based on their potential utilities. Suppose we have access to both intermediate and long-term reward estimation for candidate action sequence $a_{t:T}$, represented by $r_{t:T}$ and $g_{t:T}$, respectively. We define the TD($\lambda$)-style utility $U$ for this candidate action as follows

$$U = (1-\lambda)\sum_{n=0}^{T-t-1}\lambda^n G_{t:t+n} + \lambda^{T-t}G_{t:T}, \text{ where } G_{t:t+n} = \sum_{k=0}^{n-1}\gamma^k r_{t+k} + \gamma^n g_{t+n}, \quad (3)$$

where the decay parameter $\lambda$ determines the weights of longer horizon estimates that contribute to the final result which can help trade off the errors from dynamics predictions and value estimates. We construct a categorical distribution $\mathcal{P}$ using *softmax* for proposal selection:

$$P[i] = \frac{\exp(\xi U^i)}{\sum_j \exp(\xi U^j)}, \ \forall i \in [1, \cdots, N], \quad (4)$$

where $\xi$ denote the *softmax* temperature. Notably, M³PC requires only two prediction steps for planning at each timestep. Leveraging the bidirectional nature of Transformers and the masked autoencoding paradigm, M³PC can predict all future actions given current states and all future states given future actions in parallel. This parallel prediction capability mitigates the computational cost's linear growth with respect to the planning horizon which is commonly observed in planning algorithms such as beam search in TT (Janner et al., 2021) or CEM in TD-MPC (Hansen et al., 2022). We detail the decision-making process for reward-maximization in Algorithm 1. Since RTG value is a trajectory-wise Monte Carlo estimation, it becomes uninformative when datasets' behavior policies are diverse. We can optionally extend M³PC by replacing RTG guidance with a transition-wise value for a better heuristic. In this case, we calculate this value with a standalone value estimator updated in a dynamic programming way proposed in IQL (Kostrikov et al., 2021).

Using the 'Utility' metric to estimate future actions before they were taken, forward M$^3$PC can also adapt to an exploration strategy in the subsequent online finetuning phase, where equation 3 are used again. During the offline-to-online process, instead of executing the expectation in categorical distribution equation 4, the M$^3$PC agent samples actions from the candidate set according to the possibilities proportional to their utility. This introduces stochasticity, maintaining overall superior actions while ensuring diversity in the experience collected during exploration, thereby balancing the exploration and exploitation.

---

**Algorithm 1** Forward M$^3$PC for Reward Maximization

---

1: **Input:** Current state $s_t$, past trajectory $\tau_{<t}$, discount factor $\gamma$, decay parameter $\lambda$, number of candidates $N$, softmax temperature $\xi$
2: **Initialize:** Proposal action set $\mathcal{A}$, Utilly set $\mathcal{U}$
3: **Output:** Selected action $a$
4: $\mathcal{A} \leftarrow$ Initialize an empty list for candidate actions
5: $\alpha_{t:T} \leftarrow$ Predict uncertainty-aware action distribution sequence using [RCBC] mask as Fig. 2
6: **for** $i = 1$ to $N$ **do**
7:      $a_{t:T}^i \leftarrow$ Sample a candidate action sequence from distribution $\alpha_{t:T}$
8:      $s_{t+1:T} \leftarrow$ Roll out the candidate sequence with [FD] mask as Fig. 2
9:      $r_{t:T}^i, g_{t:T}^i \leftarrow$ Simulate intermediate rewards and long-term rewards using [RP] mask as Fig. 2
10:      $U^i \leftarrow$ Calculate expected utility                  ▷ using Equation 3
11:      Append $a_t^i, U_i$ to $\mathcal{A}, \mathcal{U}$, respectively.
12: **end for**
13: $\mathcal{P} \leftarrow$ Construct candidate selection distribution             ▷ using Equation 4
14:
15: **return** $a \leftarrow \left[\mathcal{A}^i | i \sim \mathcal{P}\right]$ **if** online, **else** $a \leftarrow \mathbb{E}_{i \sim \mathcal{P}}\left[\mathcal{A}^i\right]$

---

**Backward M$^3$PC for Goal Reaching.** The ability of a BTM to infer past tokens conditioned on future events sets it apart from GPT-based models. This feature is particularly advantageous for implementing MPC from a reverse or "backward" perspective when the objective is to achieve a specified goal state. Unlike the goal-reaching mask proposed in previous works (Liu et al., 2022; Carroll et al., 2022), which masks all elements along the trajectory except the current and final states to reconstruct the action at the current timestep, we leverage the BTM's bidirectional conditioning capability to inpaint a transition path that guides action selection. We refer to this method as backward M$^3$PC.

Specifically, the backward M$^3$PC approach uses a Path Inference (PI) mask (illustrated in Figure 3(b)) to guide the model in predicting a sequence of intermediate states leading to the goal. Once a path is established, the model employs an Inverse Dynamics (ID) mask to deduce the necessary actions to transition between consecutive states along the predicted path. This approach eliminates the need to generate a large number of candidates and roll out each one, which inherently demands significant computational resources. Instead, it implicitly performs the same function as traditional MPC by selecting the first action in a sequence that most satisfies the given goal.

## 5 EXPERIMENTS

Our experiments aim to answer the following questions:

     Q1: Can forward M$^3$PC enable the (same) agent to achieve higher accumulated rewards in offline RL and subsequent online finetuning?

     Q2: Can backward M$^3$PC enable the agent to perform diverse tasks given target states?

     Q3: How does each algorithmic component contribute to M$^3$PC?

     Q4: Is the pretrained model capable enough to perform M$^3$PC in more complex environments that demand the knowledge of interaction with external objects, e.g. manipulation?

**Tasks and Datasets.** To answer these questions, we utilize **D4RL** and **RoboMimic** dataset suites. We apply three D4RL locomotion domains (`Hopper`, `Walker2d`, `HalfCheetah`) with two

dataset types for each task: `medium(m)` and `medium-replay(m-r)`, used to benchmark our proposed forward $M^3PC$ in offline RL and O2O settings. The RoboMimic encompasses three manipulation tasks (`Can`, `Lift`, `Square`). We utilize three official datasets (`can-pair`, `square-mh`, `lift-mg`) and two customized datasets (`can-lim`, `can-real`) to evaluate $M^3PC$'s potential real-world application, particularly in robotic manipulation tasks. Detailed descriptions of the tasks and datasets are provided in Appendix D.

Table 1: **Offline Results on D4RL.** Comparison of the average normalized return against several baseline methods **without** online finetuning. $M^3PC$-M and $M^3PC$-Q are shortened for our method $M^3PC$ with (M)onte-carlo return estimation and (Q)-value estimation guidance heuristics, respectively. We report the mean and standard deviation of 5 seeds. The best result for each dataset is highlighted in **bold**. Note that $M^3PC$-M shares the same weights as a pretrained BTM, but constantly outperforms BTM in all tasks due to the test-time enhancement brought by $M^3PC$.

| Dataset | BC | TD3+BC | IQL | DT | TT | BTM | $M^3PC$-M | $M^3PC$-Q |
|---------|-----|--------|-----|-----|-----|-----|-----------|-----------|
| hopper-m | 53.5 | 60.4 | 63.8 | 65.1 | 61.1 | 64.3 | $70.7_{\pm6.2}$ | $\mathbf{73.6}_{\pm5.6}$ |
| walker2d-m | 63.2 | 82.7 | 79.9 | 67.6 | 79.0 | 72.5 | $80.9_{\pm2.5}$ | $\mathbf{86.4}_{\pm2.6}$ |
| halfcheetah-m | 42.4 | 48.1 | 47.4 | 42.2 | 46.9 | 43.0 | $43.9_{\pm3.9}$ | $\mathbf{51.2}_{\pm0.7}$ |
| hopper-m-r | 29.8 | 64.4 | **92.1** | 81.8 | 91.5 | 75.3 | $80.4_{\pm5.2}$ | $78.3_{\pm16.2}$ |
| walker2d-m-r | 21.8 | 85.6 | 73.7 | 82.1 | 82.6 | 76.6 | $78.2_{\pm10.2}$ | $\mathbf{92.2}_{\pm2.4}$ |
| halfcheetah-m-r | 35.7 | 44.8 | 44.1 | **48.3** | 41.9 | 41.1 | $41.8_{\pm0.5}$ | $48.2_{\pm0.4}$ |
| Total | 246.4 | 386.0 | 401.0 | 387.1 | 403.0 | 372.8 | 395.9 | **429.8** |

**Q1: Offline RL.** We present the offline results of $M^3PC$ with **M**onte Carlo return estimation guidance ($M^3PC$-M) and **Q**-value estimation guidance ($M^3PC$-Q) in Table 1. To evaluate the offline RL performance of our proposed method, we compare it against the following baselines: (1) BC: behavior cloning, which directly mimics the behaviors in the offline dataset; (2) TD3+BC (Fujimoto & Gu, 2021): an off-policy RL method incorporating a behavior cloning regularization term; (3) IQL (Kostrikov et al., 2021): a model-free algorithm designed to avoid bootstrapping errors by learning implicit Q-functions; (4) DT (Chen et al., 2021): a sequence-modeling model free approach that predicts actions conditioned on expected returns; (5) TT (Janner et al., 2021): a sequence-modeling model based approach that utilizes beam search planning and (6) BTM: which shares the same pretrained model as our method but applies only the [RCBC] mask for policy inference. The results demonstrate that $M^3PC$ significantly improves reward accumulation compared to BTM, consistently outperforming it across all datasets and domains, irrespective of the guidance heuristic used. This indicates that $M^3PC$'s planning phase effectively refines the action proposals generated by BTM. Furthermore, as a generalist agent, $M^3PC$-M performs competitively with specialized offline RL algorithms such as TD3+BC and IQL. Notably, $M^3PC$-Q achieves even more competitive results, outperforming all baselines by a considerable margin.

**Online Finetuning.** Under the O2O setup, we compare our method against IQL and ODT (Zheng et al., 2022), a specially designed O2O method for DT. The full online training curves of each algorithm can be found in Appendix C. In Table 2, we report the performance of each algorithm with a 200K online sample budget. To ensure a fair comparison, we use the best performance between ODT's original paper (Chen et al., 2021) and our result running its open-sourced implementation. Our method outperforms the other two methods in all the environments except the `hopper-medium` dataset. After fine-tuning, our total performance score is 31% higher than IQL and 26% higher than ODT, with improvements over finetuning 123% more substantial than those of ODT. We plot the normalized exploration rollout statistics of the $M^3PC$ agent and the BTM agent in Figure 4. The results show that $M^3PC$ is more likely to collect trajectories of high quality while maintaining some randomness to cover diverse states. Additional results are provided in Appendix C.

**Q2: Goal Reaching.** To assess whether our proposed method can effectively guide an agent to specified goal states, we evaluate it on the following three tasks: (a) Halfcheetah flipping, (b) Walker performing splits, and (c) Hopper wiggling. Due to the limited planning horizon of our model, we provide a sequence of consecutive subgoals to ensure that each goal-reaching task remains within

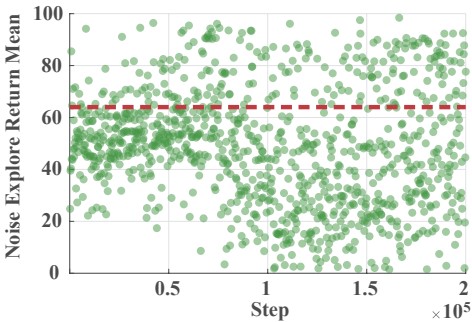
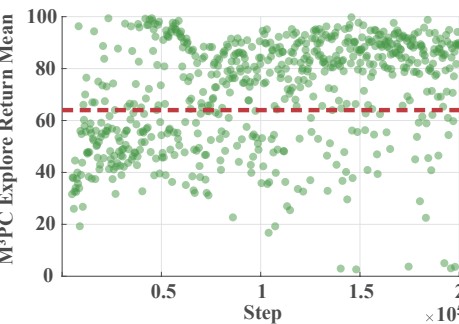

Figure 4: **Exploration Rollout Statistics.** Results from two example runs of the `Hopper` task on the `medium` dataset using the same offline pretrained BTM agent. One run employs Gaussian noise for exploration, while the other utilizes M³PC. The red line represents the offline result. Compared to naive Gaussian noise exploration, M³PC significantly improves the agent's exploration quality by generating more high-return trajectories while maintaining stochasticity, including some mid-level or low-return trajectories.

Table 2: **Online Finetuning Results on D4RL.** Comparison of normalized returns before and after online finetuning, as well as the improvement achieved using a **200K** online sample budget. We report the mean from five seeds. The best final result for each dataset is highlighted in **bold** and the greatest improvement is highlighted in green.

| Dataset | IQL | | | ODT | | | M³PC (Ours) | | |
|---|---|---|---|---|---|---|---|---|---|
| | offline | online | $\delta$ | offline | online | $\delta$ | offline | online | $\delta$ |
| hopper-m | 63.8 | 66.8 | +3.0 | 67.0 | **97.5** | +30.6 | $73.6_{\pm5.6}$ | $93.9_{\pm15.8}$ | +20.3 |
| walker2d-m | 79.9 | 80.3 | +0.4 | 72.2 | 76.8 | +4.6 | $86.4_{\pm2.6}$ | $\mathbf{91.9}_{\pm7.8}$ | +5.5 |
| halfcheetah-m | 47.4 | 47.4 | +0.0 | 42.7 | 42.2 | -0.6 | $51.2_{\pm0.7}$ | $\mathbf{69.3}_{\pm2.1}$ | +18.1 |
| hopper-m-r | 92.1 | 96.2 | +4.1 | 86.6 | 88.9 | +2.3 | $78.3_{\pm16.2}$ | $\mathbf{103.5}_{\pm6.0}$ | +25.2 |
| walker2d-m-r | 73.7 | 70.6 | -3.1 | 68.9 | 76.9 | +7.9 | $92.2_{\pm2.4}$ | $\mathbf{105.2}_{\pm1.0}$ | +13.0 |
| halfcheetah-m-r | 44.1 | 44.1 | +0.0 | 40.0 | 40.4 | +0.4 | $48.2_{\pm0.4}$ | $\mathbf{67.0}_{\pm7.1}$ | +18.8 |
| Total | 401.0 | 405.5 | +4.5 | 377.4 | 422.7 | +45.3 | 429.8 | **530.8** | +101.0 |

the model's planning horizon capacity, rather than directly providing the final desired goal state. Details regarding subgoal selection for each task are provided in Appendix A.

These tasks deviate from the reward mechanisms typically seen in offline data but can be extrapolated or stitched together from offline trajectories. Our results, showcased in Figure 5, illustrate that backward M³PC enables the agent to generalize to diverse tasks rather than merely imitating offline experiences. This demonstrates the model's ability to adapt to new challenges by leveraging its knowledge of complex dynamics to achieve specific goals.

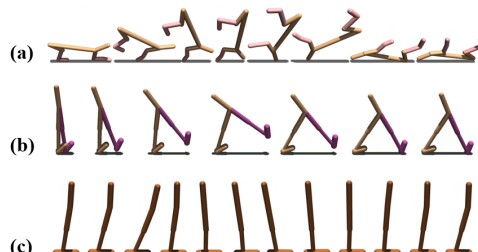

Figure 5: **Demonstration for D4RL Goal Reaching.** One evaluation visualization for (a) Halfcheetah flipping, (b) Walker doing splits, and (c) Hopper wiggling at a predefined frequency. These behavior are all unseen in the offline dataset during pretraining, see Appendix C for more details.

Additionally, we evaluated the BTM's goal-reaching ability using a single goal-reaching mask, similar to previous studies (Carroll et al., 2022; Liu et al., 2022). This method involves keeping the

current state and goal state unmasked while directly executing the inpainted action. However, as detailed in Appendix C, this approach failed to consistently enable the agent to reach the goal state. This discrepancy underscores the effectiveness of our model-based approach.

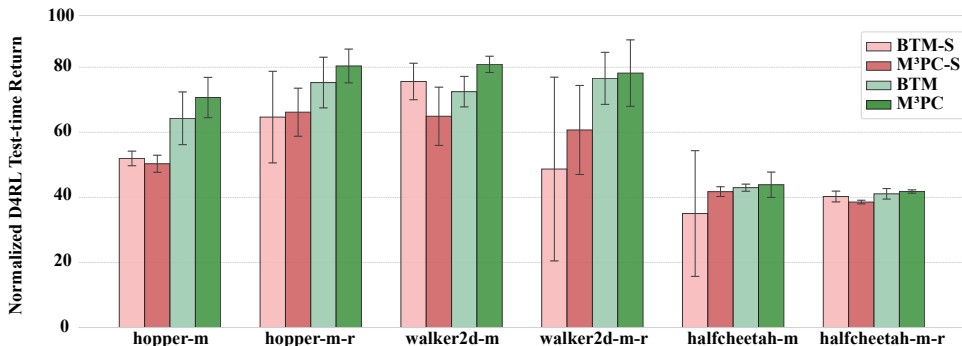

Figure 6: **Offline RL comparison between unified and specialized model.** We report the normalized average returns of BTM and M$^3$PC on a unified pretrained agent compared to specified pretrained agents, denoted by BTM-S and M$^3$PC-S, respectively. The results represent the mean over five seeds. The comparison suggests that the unified pretrained model leads to more efficient representations and better performance with BTM and M$^3$PC.

**Q3: Ablation Studies.** We conduct ablation studies to evaluate the contribution of individual components to the success of our method. Specifically, we examine whether unifying the pretraining process using a random masking technique enhances M$^3$PC performance. To this end, we pretrained two separate models—a policy model and a world model—using the same training objective and model structure as the BTM. However, these models were only applied with the [RCBC] mask and [FD] mask, respectively, during the training phase. These specialized models were then integrated to implement MPC. Our findings, illustrated in Figure 6, indicate that two specialized pretrained models do **not** improve decision quality compared to the unified pretrained BTM. Moreover, implementing MPC with separate policy and world models does not significantly enhance decision-making compared to using only the specialized policy model. This suggests that the unified pretraining approach benefits performance, as the bidirectional Transformer cohesively captures both policy behaviors and environmental dynamics, leading to more effective planning during MPC.

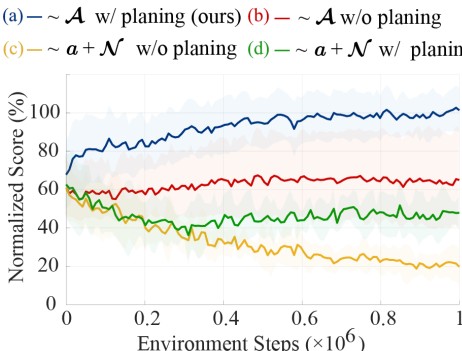

Figure 7: **Ablation Study on Planning and Uncertainty-aware Action Reconstruction.** We ablate sample-based planning, uncertainty-aware action reconstruction, and both components to investigate their contributions to the algorithmic performance in the online finetuning phase. We report average results over six datasets. Mean of five seeds. The shaded area represents the averaged per-task standard deviation across random seeds.

We further justify some design choices in M$^3$PC's online finetuning phase by comparing: (a) Our M$^3$PC as in Algorithm 1, which combines uncertainty-aware action distribution $\mathcal{A}$ reconstruction and planning-based action resampling (b) randomly sampling from $\mathcal{A}$ for exploration (c) the original BTM's method of action $a$ reconstruction, trained with MSE loss, adding fixed action noise $\mathcal{N}(\mathbf{0}, \sigma I)$ for exploration, maintaining the same entropy level as M$^3$PC. (d) performing planning-based action resample using (c)'s decisions. We present the averaged finetuning process across D4RL datasets in Figure 7. The results highlight the effectiveness of our key contributions. Specifically, an uncertainty-aware policy for exploration is crucial for maintaining online training stability,

and forward planning significantly improves sample efficiency. Figure 7 also shows that the performance drops drastically when naively using the uncertainty-"unaware" original BTM for exploration. Per-task training curves and additional ablation studies are provided in Appendix C.

Table 3: **Offline Results on RoboMimic.** Success rate of various offline pretrained agents in manipulation tasks. We report the mean of 5 seeds (50 trials for simulator and 20 trials for real world). We exclude the BC and IQL from real-world implementation due to their poor performance in the corresponding simulated tasks.

| Dataset | BC | IQL | DT | $M^3PC$ |
|---|---|---|---|---|
| Can-Pair | 0.64 | 0.34 | 0.94 | **0.98**$_{\pm 0.01}$ |
| Square-MH | **0.53** | 0.13 | 0.21 | 0.28$_{\pm 0.14}$ |
| Lift-MG | 0.65 | 0.29 | **0.93** | 0.77$_{\pm 0.07}$ |
| Can-Lim | 0.25 | 0.27 | 0.46 | **0.54**$_{\pm 0.16}$ |
| Can-Real | - | - | 0.50 | **0.70**$_{\pm 0.10}$ |

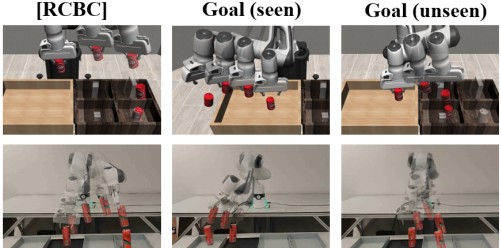

Figure 8: **Skill Generalization in Can-Pick task.** Simulated environments on the top and real-world environments on the bottom. The columns show the original behavior (left), behavior conditioned on the seen goal state (mid), and behavior conditioned on the unseen goal state (right).

**Q4: Manipulation.** In addition to the self-body motion control tasks explored in earlier experiments, we shift our focus to manipulation tasks to assess whether the proposed $M^3PC$ method can be effectively applied to robotics tasks requiring interaction with objects in the environment. We conducted experiments across three simulated tasks in RoboMimic—Can, Square, and Lift—each with varying levels of complexity. Additionally, we utilized datasets of varying quality levels, including machine-generated (MG), mid-level human-demonstrated (MH), and paired positive-and-negative (Pair) demonstrations. We also created a customized simulated task named Can-Lim, a variant of the Can-Pick task, in which the dataset was adapted to a scenario where the relative pose between the gripper and the can is unavailable. Finally, we tested our method on a real-world Can-Pick task, referred to as Can-Real. The results are compared against several offline RL baselines, as shown in Table 3.

To evaluate generalization capabilities, we conducted a goal-conditioned RL experiment on the Can-Picking task with the Paired dataset. This dataset contains 50% perfect demonstrations, where the agent successfully picks up the can and places it into the box in the right corner, and 50% negative demonstrations, where the can is thrown off the table, resulting in no reward. As illustrated in Figure 8, by specifying the final goal states, we can control the agent's behavior to either complete the original task or reproduce the throwing-away behavior. Furthermore, by specifying a final state numerically between two observed states in the dataset, the model generates actions that enable the agent to reach a previously unseen state—placing the can into the box adjacent to the right one.

## 6 DISCUSSIONS AND LIMITATIONS

We propose $M^3PC$, a test-time MPC framework designed to enhance the inference performance of masked Transformers pretrained under offline RL settings. $M^3PC$ offers the following benefits: **(1) Improved Decision-Making without Further Training**: During inference, $M^3PC$ improves decision-making with high computational efficiency. **(2) Enhanced Finetuning Efficiency:** With an additional online interaction budget, $M^3PC$ achieves better final performance and outperforms the previous sequential modeling O2O approach, ODT. This enhances the agent's continuous learning ability. **(3) Generalization Ability:** The framework demonstrates notable generalization capabilities, generating actions that effectively drive the agent toward unseen goal states in both simulated and real-world tasks.

While $M^3PC$ demonstrates promising results, there are areas for further investigation: **(1) Handling Pixel Observations:** Currently, our framework is limited to environments with state-based observations. Future work will explore methods for handling pixel observations. **(2) Transformer Scalability:** Our experiments employed a fixed-structure masked trajectory Transformer. It remains unclear whether increasing the Transformer's capacity would lead to better test-time planning.

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

## A   IMPLEMENTATION DETAILS

**Loss Function Construction.**  We consider the Lagrangian of Equation 2 given by:

$$L(\theta, \sigma) = J(\theta) + \sigma(\beta - \mathcal{H}_\theta^\mathcal{T}), \tag{5}$$

where $\sigma$ is a non-negative Lagrange multiplier. The training objective then become

$$\max_{\sigma \geq 0} \min_\theta L(\theta, \sigma). \tag{6}$$

We alternately optimize $\theta$ and $\sigma$ as follows:

- **Optimizing $\theta$ with fixed $\sigma$**, which involves:

$$\min_\theta \left( J(\theta) - \sigma \mathcal{H}_\theta^\mathcal{T} \right), \tag{7}$$

- **Optimizing $\sigma$ with fixed $\theta$**, formulated as:

$$\min_{\sigma \geq 0} \sigma \left( \mathcal{H}_\theta^\mathcal{T} - \beta \right). \tag{8}$$

This iterative training of $\theta$ and $\sigma$ ensures compliance with the entropy constraint while optimizing the objective function $J(\theta)$.

**Transition-wise Value Estimator.**  We choose IQL (Kostrikov et al., 2021) algorithm to train the value estimator because its Bellman updates do not require an explicit policy function. Typically, IQL simultaneously learns a critic network $Q_\psi$ and value network $V_\phi$ with the losses defined by:

$$
\begin{aligned}
J_\mathcal{Q}(\psi) &= \mathbb{E}_{(s,a,r,s')\sim\mathcal{T}} \left[ \left( r + \gamma V_\phi(s') - Q_\psi(s,a) \right)^2 \right], \\
J_V(\phi) &= \mathbb{E}_{(s,a)\sim\mathcal{T}} \left[ \left| \mathbb{t} - \mathbb{1}_{\{\mathcal{Q}_\psi(s,a) - V_\phi(s) < 0\}} \right| \left( \mathcal{Q}_\psi(s,a) - V_\phi(s) \right)^2 \right]
\end{aligned}
\tag{9}
$$

, where $\mathbb{t}$ is a constant hyperparameter named *expectile* used to control the conservatism of the value estimation. The critic network $Q_\psi$ will be applied to estimate the long-term reward for a given state-action pair in our approach. $\mathbb{t}$ is set to 0.7 for D4RL locomotion tasks and 0.9 for RoboMimic manipulation tasks.

**Goal State Definitions in Goal Reaching Tasks.**  In the goal-reaching setup for Hopper, Walker, and HalfCheetah, we craft rough trajectories based on the specific anticipated dynamics of each agent. For the Hopper, a sinusoidal trajectory is designed for the foot joint to induce a wiggling motion, while the other two joints' initial positions are maintained. In the case of the Walker, a linearly increasing trajectory for the thigh joint facilitates the splits, with the dynamics of other joints extracted from the offline trajectory which correspond to a stepping behavior, providing rough guidance. For the HalfCheetah, the primary flipping motion is guided by linear trajectories that set the body height decrease and a full 180-degree rotation to simulate a flip. Complementing this, the dynamics of the other joints and body movements are derived from offline datasets that capture detailed flipping steps, providing a coherent and realistic motion base. Subgoals are extracted from these trajectories at specific intervals—every fifth, thirtieth, and every timestep, respectively—to guide each model towards achieving the intended maneuvers, ensuring that while the main actions are precisely targeted, the full spectrum of body dynamics remains realistically integrated and synchronized with the models' overall movements.

In the Can Pick task, we deploy specific guidance trajectories for each distinct behavior—throwing away and moving to a nearer box. For the "throwing-away" behavior, we directly select a suitable trajectory from an offline dataset without any modifications, ensuring that the agent replicates a proven effective throwing motion. For the "moving to a nearer box" behavior, the process begins by selecting a "moving to correct box" trajectory from the offline dataset. To tailor this trajectory to the specific task, we apply an affine transformation to adjust the horizontal positions of the Franka robot's end effector and the object along the trajectory. This transformation proportionally reduces the distance the object needs to be moved, customizing the trajectory to the current scenario. Subgoals are then extracted from these guidance trajectories at every state, providing detailed, step-by-step targets that guide the agent's actions towards successful task completion.

**Hardware.**  The entire training process, including both pretraining and finetuning, is performed on NVIDIA 3090 GPUs. During the offline pretraining phase, we train the BTM model for 140K

gradient steps, which takes approximately 4 hours per dataset on a single GPU. For the finetuning phase, we allow 1 million online exploration steps for figure plot and report the performance with 0.2 million exploration steps. The finetuning phase including exploration and evaluation in simulator takes between 7 and 9 hours per dataset on a single GPU, while finetuning the pretrained trajectory model itself takes half of the total time.

## B    HYPERPARAMETERS

Table 4: **Hyperparameters.**

| Hyperparameter | Offline | Online |
|---|---|---|
| **Training** | | |
| Nonlinearity | GELU | GELU |
| Batch size | 2048 | 512 |
| Trajectory-segment length | 8 | 8 |
| Dropout | 0.10 | 0.10 |
| Learning rate | 0.0001 | 0.0001 |
| Weight decay | 0.005 | 0.005 |
| Target entropy $\beta$ | -3 | -3 |
| Scheduler | cosine decay | - |
| Warmup steps | 40000 | - |
| Training steps | 140000 | - |
| **Evaluation** | | |
| Context length | 4 | 4 |
| **Bidirectional Transformer** | | |
| # of Encoder Layers | 2 | 2 |
| # of Decoder Layers | 1 | 1 |
| # Heads | 4 | 4 |
| Embedding Dim | 512 | 512 |
| **Mode Decoding Head** | | |
| Number of Layers | 2 | 2 |
| Embedding Dim | 512 | 512 |
| **Reward Maximization** | | |
| Decay Parameter $\lambda$ | 0.6 | 0.6 |
| Candidate Number $N$ | 625 | 625 |
| Softmax temperature $\xi$ | 1.0 | 1.0 |

# C    ADDITIONAL RESULTS

**Online Finetuning Results.** We report the per-task online training curves over 1 million online samples for our method and our reproductions of baseline methods in Figure 9, ablations in Figure 10.

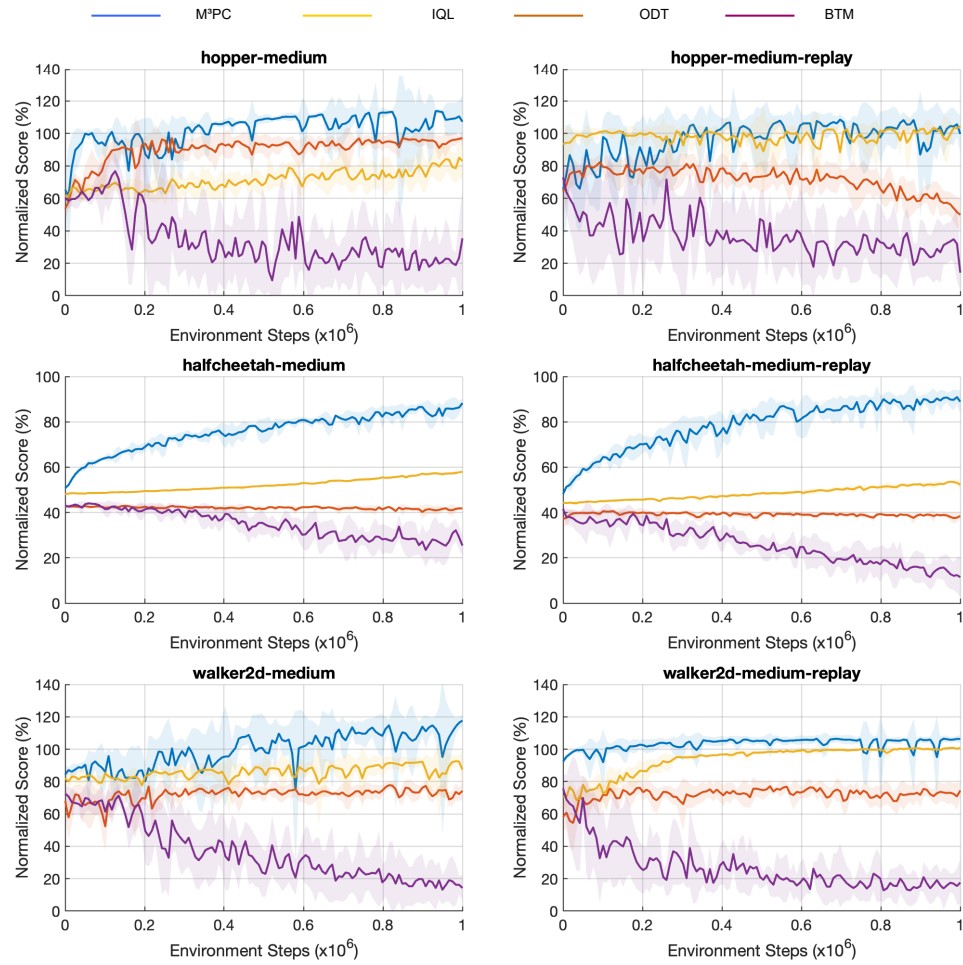

Figure 9: **D4RL Benchmark Comparison.** Per-task Online Training Curves for $M^3PC$ and baseline methods. Mean of 5 seeds. The shaded area represents the standard deviation across seeds.

We furthermore compete $M^3PC$ with some stronger, specialized O2O baseline methods with the 100k online sample budget practice: (1) AWAC (Nair et al., 2020), a representative O2O approach utilizing advantage-weighted actor-critic; (2) ODT (Zheng et al., 2022), a unified sequential modeling framework for offline RL and online finetuning; (3) OFF2ON Lee et al. (2022), a CQL-based pessimistic Q-ensemble method that incorporates a balanced replay to encourage near on-policy samples from the offline dataset; and (4) PEX (Kostrikov et al., 2021), an IQL-based algorithm focused on policy expansion. We evaluate the baselines on the D4RL locomotion datasets, with the results summarized in Table 5. The results demonstrate that $M^3PC$ achieves performance comparable to SOTA specialized O2O methods such as OFF2ON and PEX.

| Dataset | AWAC | ODT | OFF2ON | PEX | M³PC |
|---|---|---|---|---|---|
| hopper-m | $57.8 \to 55.1$ | $73.4 \to 67.0$ | $97.5 \to 80.2$ | $56.5 \to 87.5$ | $73.6 \to 81.3$ |
| walker2d-m | $35.9 \to 72.1$ | $72.0 \to 72.2$ | $66.2 \to 72.4$ | $80.1 \to 92.3$ | $86.4 \to 74.9$ |
| halfCheetah-m | $43.0 \to 42.4$ | $42.7 \to 42.1$ | $39.3 \to 59.6$ | $50.8 \to 60.9$ | $51.2 \to 64.0$ |
| hopper-mr | $37.7 \to 60.1$ | $60.4 \to 78.5$ | $28.2 \to 79.5$ | $31.5 \to 97.1$ | $78.3 \to 78.6$ |
| walker2d-mr | $24.5 \to 79.8$ | $44.2 \to 71.8$ | $17.7 \to 89.2$ | $80.1 \to 92.3$ | $92.2 \to 98.8$ |
| halfCheetah-mr | $40.5 \to 41.2$ | $32.4 \to 39.7$ | $42.1 \to 60.0$ | $45.5 \to 51.3$ | $48.2 \to 62.7$ |
| Average | $39.9 \to 58.5$ | $54.2 \to 61.9$ | $48.5 \to 73.5$ | $57.4 \to 80.2$ | $71.7 \to 76.8$ |

Table 5: **O2O Baseline Comparison Results.** Comparison of normalized returns before and after online finetuning with a **100K** online sample budget. We report the mean of four seeds.

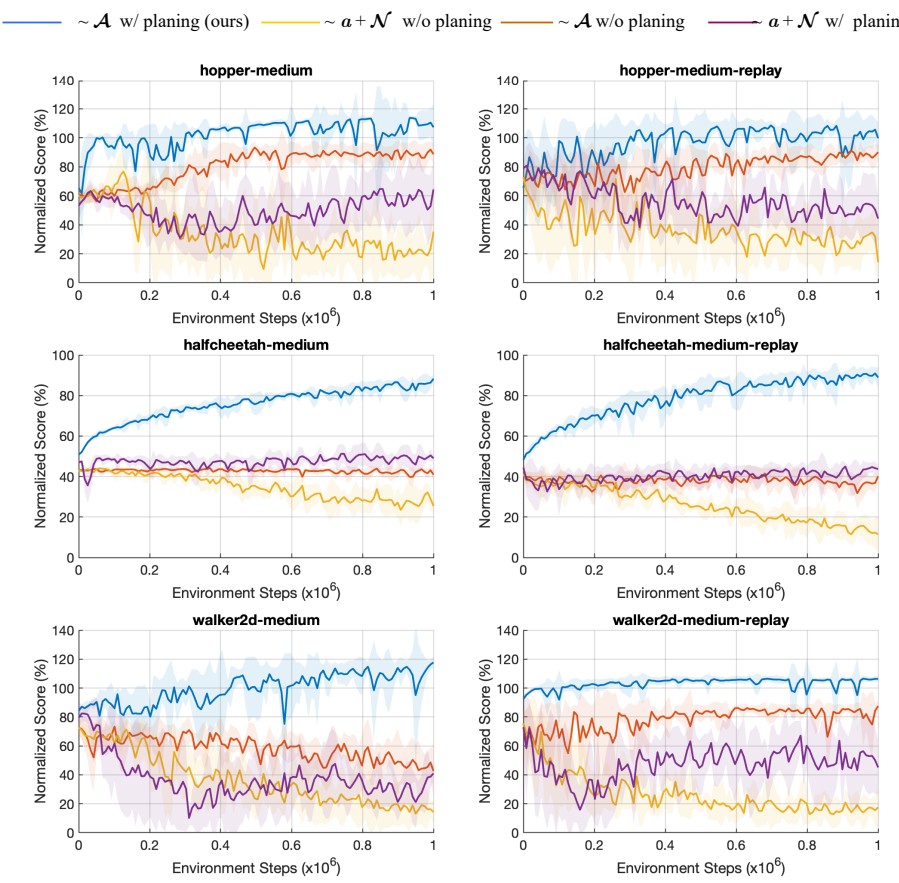

Figure 10: **Ablation Studies for Algorithmic Components Contribution.** Mean of 5 seeds. The shaded area represents the standard deviation across seeds.

**Inference Time.** We have introduced M³PC's computational efficiency due to the parallel prediction nature of the mask autoencoding paradigm in the methodology section. For completeness, we report the inference time of M³PC's planning overhead with respect to a range of planning horizons (1 to 8) in Fig. 11. We additionally include two methods for references: (1) TT (Janner et al., 2021), a sequential modeling approach that employs beam search for test-time planning; (2) TD-MPC (Hansen et al., 2022), a representative model-based RL method combining MPC and temporal difference learning. All inference times were benchmarked on a single NVIDIA RTX 3090 GPU.

Note that we used the original implementations of the baseline methods, so the number of parameters is not aligned across approaches. Results demonstrate that M³PC is much more computational efficient compared to the sequential modeling approach TT. Furthermore, as the planning horizon increases, M³PC even outperforms TD-MPC, despite the latter being a more lightweight model.

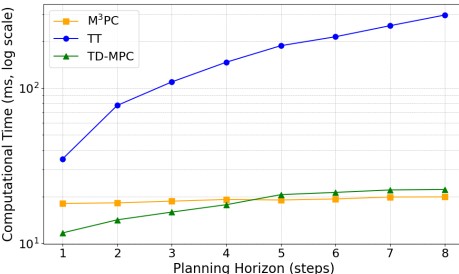

Figure 11: **Inference Time Comparison.** M³PC is much more computational efficient compared to sequential modeling approach TT and even outperform lightweight model TD-MPC as planning horizon increases.

**Ablation Study on Decay Parameter.** Decay parameter $\lambda$ play a significant role in balancing the weight of instant rewards and long-term value. We provide the training curves for $\lambda \in \{0.0, 0.1, 0.3, 0.5, 0.7, 0.9\}$. Figure 12 indicates our approach is not sensitive to the choice for $\lambda$ since each choice outperforms the baseline (randomly sampling action from $\mathcal{A}$ for exploration) by a large margin, and has minor difference in learning speed (fine-tuning improvements happen slower when $\lambda = 0.1$ and long step stability (performance drops after 800k online steps when $\lambda = 0.9$. We choose $\lambda = 0.6$ in all the experiments as an intermediate choice for balancing converge speed and online training stability.

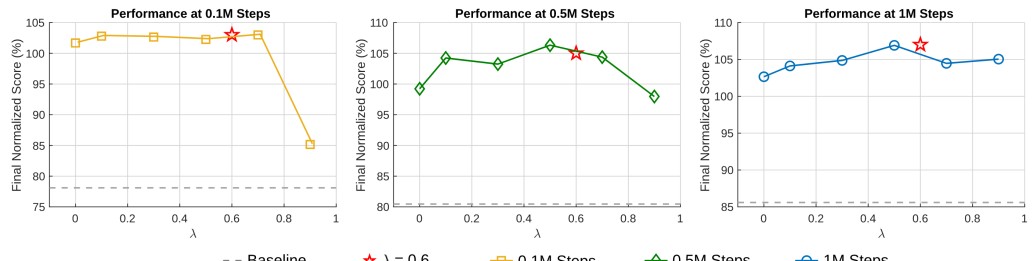

Figure 12: **Ablation Study for $\lambda$ Choices.** Normalized score as a function of $\lambda$ choice with 0.1m, 0.5m, 1.0m online steps. The red star represents our default choice (0.6) while the grey line denotes baseline results (explore w/o planning). Mean of 3 seeds.

**Ablation Study on Entropy Constraint.** We also report the effects of entropy constraint we imposed in Equation 2. The results of offline results M³PC-M, M³PC-Q and online finetuning results M³PC-online are summarized in Table 6. Empirical results show that entropy constraint does not have substantial influences on offline results but significantly boost the online sample efficiency.

| Datasets | M³PC-M | | M³PC-Q | | M³PC-online | |
|---|---|---|---|---|---|---|
| | w/o | w | w/o | w | w/o | w |
| hopper-m | $84.3_{\pm7.3}$ | $70.7_{\pm6.2}$ | $81.6_{\pm3.5}$ | $73.6_{\pm5.6}$ | $94.9_{\pm11.7}$ | $93.9_{\pm15.8}$ |
| halfcheetah-m | $43.8_{\pm0.6}$ | $43.9_{\pm3.9}$ | $50.0_{\pm0.3}$ | $51.2_{\pm0.7}$ | $71.5_{\pm3.6}$ | $69.3_{\pm2.1}$ |
| walker2d-m | $79.9_{\pm1.4}$ | $80.9_{\pm2.5}$ | $80.7_{\pm7.2}$ | $86.4_{\pm2.6}$ | $68.3_{\pm25.0}$ | $91.9_{\pm7.8}$ |
| hopper-mr | $75.1_{\pm11.3}$ | $80.4_{\pm5.2}$ | $76.8_{\pm27.2}$ | $78.3_{\pm16.2}$ | $88.7_{\pm26.9}$ | $103.5_{\pm6.0}$ |
| walker2d-mr | $78.5_{\pm16.0}$ | $78.2_{\pm10.2}$ | $94.0_{\pm0.8}$ | $92.2_{\pm2.4}$ | $108.1_{\pm3.5}$ | $105.2_{\pm1.0}$ |
| halfcheetah-mr | $40.0_{\pm1.0}$ | $41.8_{\pm0.5}$ | $48.0_{\pm0.8}$ | $48.2_{\pm0.4}$ | $70.2_{\pm2.8}$ | $67.0_{\pm7.1}$ |
| Average | 66.9 | 66.0 | 71.8 | 71.6 | 83.6 | 88.5 |

Table 6: **Ablation Study on Entropy Constraint.** Comparison of M³PC-M, M³PC-Q, and online results w or w/o entropy constraint across D4RL datasets.

**Goal Reaching Results.** We show more results in goal reaching tasks here. To demonstrate the extent to which our unseen goal is out of the distribution, we show together the PCA dimension-reduced results for all states in the offline dataset on which the model was pretrained, and the states in the trajectory of reaching the given goal. As in Fig.13, The different tasks have different out-of-distribution cases: For the walker-split task, the agent starts with a seen state and finally reach to a state never seen before (the angle of the hip-joint). For the cheetah-flip task, the initial state and goal state are both seen in the offline dataset, the normal state usually corresponds to better rewards, while the flip-over state hardly leads to any reward, as the original task in the dataset is run fast. However, conditioned on the state given, the agent finds many unseen intermediate states to finally transit to a flip-over state. For the Hopper-Wiggle task, the agent strings together a series of near-in-distribution states to form a loop of wiggling action, which is not seen in the dataset.

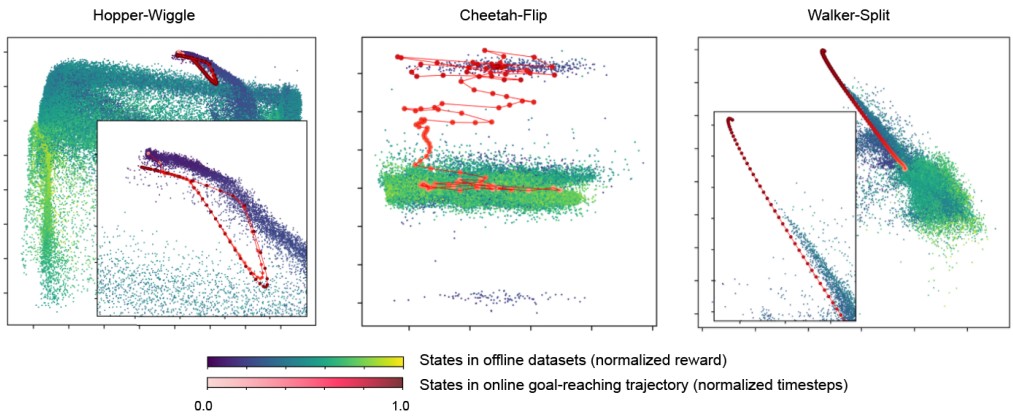

Figure 13: **Visualization of states in different tasks after 2-dim PCA mapping.**

Additionally, we show the goal states we take as input in order to reach the final behavior, and how well BTM with a single Goal Reaching mask and backward M³PC can follow those states. We only plot the most representative dimension in the state vector for each task, respectively. E.g., Angle of the front tip (dim[1]) of cheetah, and angle of the thigh joint (dim[2]) of walker and angle of the top (dim[1]) of hopper. As in Fig.14, with only a single mask, the agent can hardly achieve the goal, and the overall behavior resembles the behavior cloning result from the pretrain dataset. However, with backward M³PC, the agent can successfully follow the kinematics guidance, although some do not exactly satisfy the dynamics. Moreover, we show that the same pretrained model with backward M³PC can reach wiggling behavior of different frequencies in hopper environment, with proper goal states.

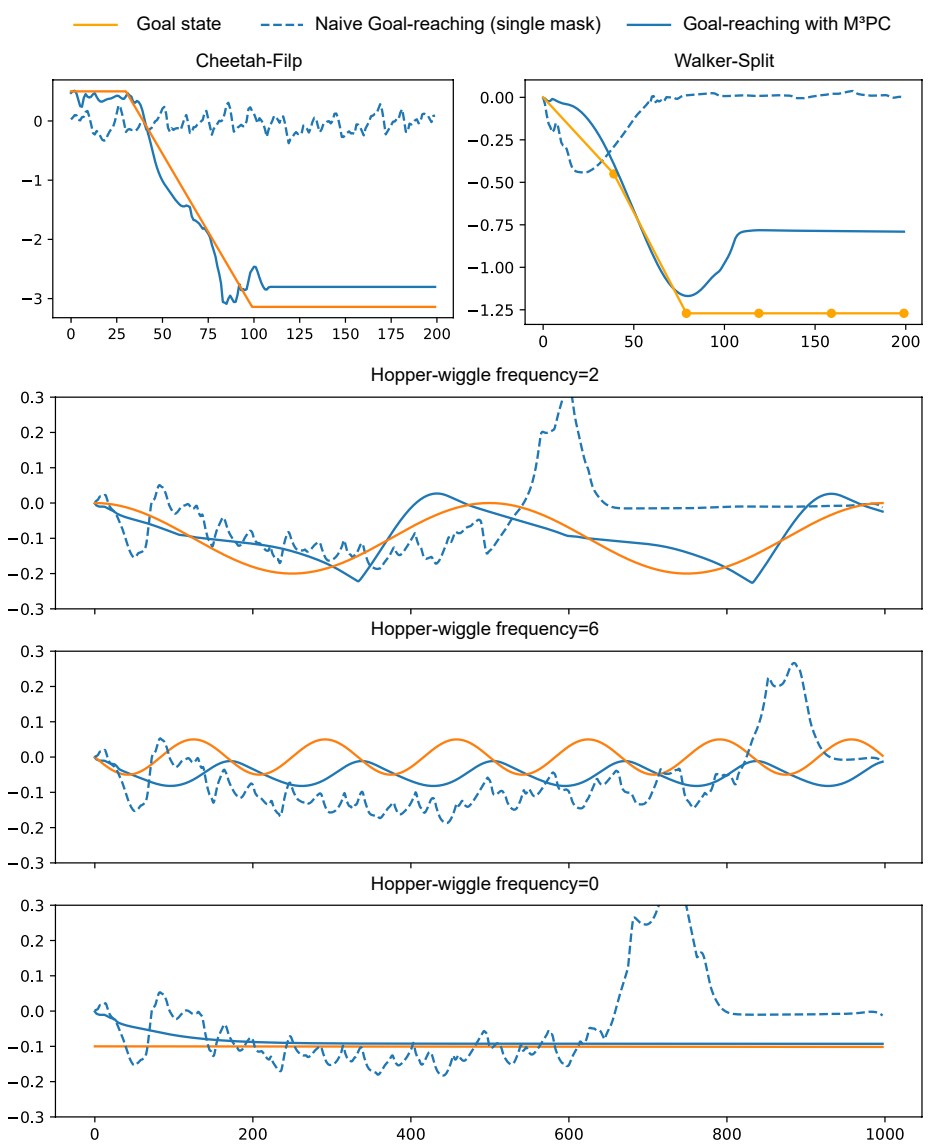

Figure 14: **Comparison between Backward M³PC and a single Mask in Goal-Reaching Tasks.**
We present the goal states and resulting states after policy execution across three goal-reaching tasks,
focusing on a single key dimension. The single Mask fails to guide the agent toward the goal states
when the given current-goal state pairs are out of distribution.

# D  TASKS AND DATASETS

The dataset utilization checklist is shown in Table 7.

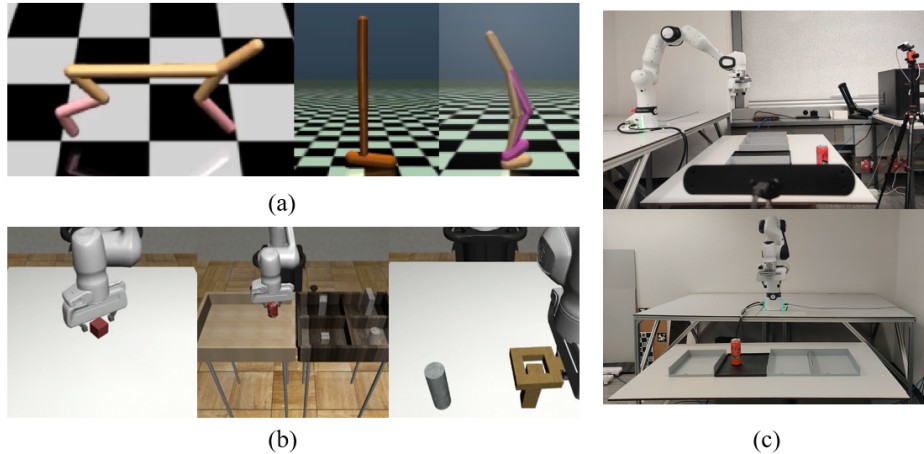

Figure 15: **Tasks Setup. (a)** Locomotion tasks in D4RL: halfcheetah, hopper, walker2d (from left to right); **(b)** Manipulation tasks in RoboMimic: lift, can, square (from left to right), **(c)** Left view and front view of real-world manipulation task setup.

**D4RL.** We consider three representative D4RL locomotion domains (`Hopper`, `Walker`, and `HalfCheetah`). Each domain contains two datasets (`medium`, `medium-replay`) which have different data compositions. The `medium` datasets contain 1M samples collected by a partially-trained SAC (Haarnoja et al., 2018) agent. The `medium-replay` dataset consists of recording all samples in the replay buffer observed during training until the agent reaches the "medium" level. We use both these two types of datasets in offline RL and O2O RL.

**RoboMimic.** RoboMimic includes a suite of manipulation task datasets designed for the Franka Panda robot, focusing on three specific tasks: `Can`, `Square`, and `Lift`. The dataset for pretraining encompasses four distinct categories: (1) Multi-Human (MH), consisting of six sets with each containing 50 demonstrations by different pairs of demonstrators; (2) Machine Generated (MG), generated by a Soft Actor-Critic (SAC) agent at various stages of its training, providing a spectrum of behaviors from early exploratory to more refined tactics; and (3) Paired, where a single experienced operator recorded two demonstrations for each of 100 initializations of the Can task—one demonstrating correct placement and the other tossing the object outside. We detailed the state space and action space definition for each environment in Robomimic below, including our customized environments can-limit and can-real.

**The Action Space and State Space for Manipulation.** The action space for each timestep is a 7-dimensional vector per arm, where the first six coordinates represent control signals in the operational space control (OSC) space, and the last coordinate controls the opening and closing of the gripper fingers. The observation space includes a 7-dimensional vector for the absolute end effector position quaternion and a 2-dimensional vector for the left and right finger relative poses of the gripper in addition to task-specified object observations. In the "Lift" task, object observations include a 10-dimensional vector consisting of the absolute cube position and quaternion (7-dim), and the cube position relative to the robot end effector (3-dim). In the "Can" task, the object observations are a 14-dimensional vector, including the absolute can position and quaternion (7-dim), and the can's position and quaternion relative to the robot end effector (7-dim). For the "Square" task, object observations also form a 14-dimensional vector with the absolute square nut position and quaternion (7-dim) and their relative positions and quaternions (7-dim) to the robot end effector. In the "Can-Limit" task, the object observations include only the absolute can position (3-dim), excluding relative position knowledge to align with goal-reaching tasks where precise relative poses are unnecessary. In the "Can-real" task, which is a real-world environment similar to Can-Limit, object position is detected using two vertically placed depth cameras, with actions output at 20 Hz, and

robot joint torques adjusted at 500 Hz to achieve the desired Cartesian poses based on the operational space controller.

Table 7: **Dataset Utilization.** We outline the dataset utilization for each experiment part here, a checkmark means the corresponding dataset is use for pretraining.

| Dataset | Offline RL | Goal Reaching RL | Online Finetuning |
|---|---|---|---|
| `hopper-medium-v2` | ✓ | ✓ | ✓ |
| `hopper-medium-replay-v2` | ✓ | | ✓ |
| `walker2d-medium-v2` | ✓ | ✓ | ✓ |
| `walker2d-medium-replay-v2` | ✓ | | ✓ |
| `halfcheetah-medium-v2` | ✓ | | ✓ |
| `halfcheetah-medium-replay-v2` | ✓ | ✓ | ✓ |
| `Can-Pair` | ✓ | | |
| `Square-MH` | ✓ | | |
| `Lift-MG` | ✓ | | |
| `Can-Lim` | ✓ | ✓ | |
| `Can-Real` | ✓ | ✓ | |

