# OpenReview forum: "M^3PC: Test-time Model Predictive Control using Pretrained Masked Trajectory Model"
_ICLR.cc/2025/Conference — ICLR 2025 Poster_

### Official Review · Reviewer_6kkf · 2024-10-18

**Soundness:** 2
**Presentation:** 3
**Contribution:** 2
**Rating:** 6
**Confidence:** 3

**Summary:**

This paper proposes a model-based planning method M$^3$PC for offline RL. M$^3$PC has two parts — a bi-directional trajectory model and model-predictive control (MPC). The trajectory model acts as a prior policy and an environment model, and test-time MPC incorporates the trajectory model to decide the most promising actions. The novelty of this paper mainly lies in the trajectory model, which predicts actions, states, and rewards using different masks. In these experiments, it shows good results in D4RL benchmarks, offline-to-online settings, and goal-reaching tasks.

**Strengths:**

1. This paper proposes four interesting masks for a bi-directional transformer, contributing to formulating offline RL as a sequence modeling problem.

2. The presentation, with nice figures, is clear and informative.

**Weaknesses:**

This paper appears to have fewer novel contributions, and the experiments should be expanded. More details are provided in the Questions section.

**Questions:**

1. There is a typo in the caption of Figure 2: "the model can show have multiple capabilities..."

2. The legend in Figure 7 is unclear. Could the authors link the curves with methods (a), (b), (c), and (d) in the text?

3. The motivation for the constraint in Eq. (2) is not adequately explained. The trajectory-level entropy constraint could lead the policy to sample out-of-distribution actions, which is problematic in offline RL and may harm performance. It would be better to compare and explain the results for a version without the trajectory-level entropy.

4. In the related work, some model-based offline planning methods have not been discussed, such as MBOP (Argenson and Dulac-Arnold, 2021), MOPP (Zhan et al., 2022), and GOPlan (Wang et al., 2024). Notably, MOPP and GOPlan consider uncertainty during planning and tend to remove/prune trajectories with high uncertainty.

5. It would be useful to report the computation time for M$^3$PC. Typically, test-time planning methods require more computation than actor-critic methods, such as TD3-BC and IQL. Could the authors also add test-time planning baselines to Table 1?

6. In the ablation study, the training details of the policy model and the world model should be provided. Q3 shows that a unified model performs better than the separated models, but it would be better to further demonstrate that the proposed unified model outperforms those in other papers. For example, the well-known world model RSSM (Hafner et al., 2019) could be compared.

**References**

Argenson, A., \& Dulac-Arnold, G. (2021). Model-Based Offline Planning. ICLR.

Hafner, D., Lillicrap, T., Fischer, I., Villegas, R., Ha, D., Lee, H., \& Davidson, J. (2019). Learning Latent Dynamics for Planning from Pixels. ICML.

Wang, M., Yang, R., Chen, X., Sun, H., Fang, M., \& Montana, G. (2024). GOPlan: Goal-conditioned Offline Reinforcement Learning by Planning with Learned Models. TMLR.

Zhan, X., Zhu, X., \& Xu, H. (2022). Model-Based Offline Planning with Trajectory Pruning. IJCAI.

---

### Official Review · Reviewer_CMiL · 2024-10-30

**Soundness:** 4
**Presentation:** 3
**Contribution:** 4
**Rating:** 8
**Confidence:** 4

**Summary:**

The paper introduces M3PC, an approach that builds on bidirectional trajectory models trained with masked prediction. During inference, M3PC refines behavior using Model Predictive Control (MPC) applied to a pre-trained model. To support this, the model includes probabilistic action "heads," allowing it to sample actions with uncertainty awareness. This setup enables forward MPC for reward maximization and reverse MPC for goal-reaching. In forward MPC, M3PC uses a utility function (a TD($\lambda$)-like combination of local rewards and either Return-To-Go or Q-values) alongside the model's forward predictions to estimate the performance of an action sequence and propose refinements. In reverse MPC, the model first plans a trajectory toward the goal, then selects actions that follow this trajectory using inverse dynamics.

A robust set of experiments demonstrates M3PC's effectiveness across standard offline reinforcement learning, offline to online settings, goal-reaching tasks, and a real robot experiment.

**Strengths:**

The paper stands out due to its simplicity and clever conceptual foundation, building effectively on a straightforward masked trajectory approach. The evaluations are thorough and demonstrate strong performance across various benchmarks, showcasing the approach's applicability to a wide set of problems. Additionally, the paper includes helpful ablation studies that validate the importance of key design choices.

**Weaknesses:**

The only weaknesses are some missing clarity, detail, and context in the experiment section, see the list below

- Which task is used in Figure 7? It seems a bit like there is an "introduction" paragraph missing before the one starting in l451
- The acronyms MPC-M and MPC-Q) appear a bit out of nowhere, I think I could piece it together, but introducing the two variants more explicitly would be helpful
- Missing standard deviations in table 3
- Include (or replace IQL with) a "conservative Q-Learning" approach tailored to O2O for better context, (e.g. Cal-QL: Calibrated Offline RL Pre-Training for Efficient Online Fine-Tuning, Nakamoto et al 2023). Such approaches could also be included in the related work and given the standardized benchmark, results can be simply taken from the respective works.
- ( I realize that this is a bit of a personal preference / due to my background so this is just a suggestion, not affecting my assessment: I would appreciate highlighting the robo-mimic results, in particular those including a real robot, and more discussion with regards to the effect of the different datasets used)

**Questions:**

- While reading the main part of the paper I assumed the "goal reaching" task only needs the final position but from the appendix it seems there are entity goal trajectories? I guess it makes sense given the limited horizon of the model, but I would appreciate a bit more elaboration clarity here.

- How are the actions selected from predicting the inverse dynamics during backward planning (sampling or mean of the gaussian head)?

- What is indicated by the shaded areas in the reward curves?

---

### Official Review · Reviewer_HJV8 · 2024-11-03

**Soundness:** 3
**Presentation:** 3
**Contribution:** 3
**Rating:** 8
**Confidence:** 3

**Summary:**

This paper introduces a unified trajectory model (Bidirectional Trajectory Model, BTM) trained through multiple auxiliary tasks. The pretrained BTM can serve both as a policy to output actions and as an environment model to predict future states. Building on this, the authors enhance policy capabilities using an MPC-based approach. The experiments demonstrate that the proposed algorithm significantly improves performance in offline RL and effectively handles offline-to-online RL and goal-conditioned RL tasks.

**Strengths:**

1. This paper presents a versatile pretrained BTM that achieves four functionalities by designing different masking methods. The authors cleverly combine these functionalities to implement MPC and goal-reaching based on BTM.
2. M$^3$PC shows substantial improvement over previous methods (DT, BTM, and ODT) in experiments. The authors also validate the method on a real robotic arm manipulation task, greatly enhancing the paper's significance.

**Weaknesses:**

I don't have major concerns. However, I would be grateful if the authors could clarify the following questions:

1. How much additional time overhead does the MPC process introduce? Including a description of time overhead in the experiments would improve the comprehensiveness of the method evaluation.
2. It would be beneficial to test the goal-reaching capability in the antmaze task, as it may better visualize the trajectories inferred by the model in these tasks.

**Questions:**

See weaknesses.

---

### Official Review · Reviewer_cxyR · 2024-11-03

**Soundness:** 2
**Presentation:** 3
**Contribution:** 3
**Rating:** 6
**Confidence:** 3

**Summary:**

This paper introduces M^3PC, an approach that leverages properly designed masking schemes to perform test-time MPC with masked trajectory models for decision making tasks. The proposed method enables action reconstruction with uncertainties for better robustness, as well as forward and backward prediction through different masking patterns for solving various downstream tasks. Evaluations are performed on both simulated motion control and real-robot manipulation tasks, across offline and offline-to-online setups. The efficacy and generalization capabilities of the method are supported by the extensive experimental results.

**Strengths:**

1. This work proposes a novel framework that combines the pre-trained Masked Trajectory Model with MPC. Allowing action reconstruction with uncertainty improves the robustness under stochastic settings.
2. The authors provide extensive evaluations in both simulated and real-robot setups, demonstrating the effectiveness of the proposed method.
3. The authors thoroughly discussed the limitations of the proposed method.
4. This paper is well-written and easy to follow.

**Weaknesses:**

1. Perform sampling-based planning with large sequence models can usually be computationally intensive and time-consuming, given that during test time M^3PC only executes the first action of the plan in every interaction step.
2. Trajectory Transformer (TT) [1] also performs planning using sequence models, even though it does not leverage any masking schemes or bidirectional transformer, including TT as one of the baselines would be a great bonus, especially when TT also proposes to leverage the Q function learned by IQL as return heuristics.
3. Please see the first question below.

[1] Janner et al. Offline Reinforcement Learning as One Big Sequence Modeling Problem. NeurIPS 2021.

**Questions:**

1. Given that one benefit of utilizing sequence models for decision-making is their strong capability to model long-term dependencies, when performing backward M^3PC for goal-reaching tasks, I wonder how the method should be properly used in long-horizon cases where the number of state transitions needed to reach the goal state might be larger than the planning horizon/time budget of the sequence model. I also wonder if it’s possible to have some quantitative results on how the proposed method performs on one or two long-horizon goal-reaching tasks (e.g. AntMaze-Large).
2. The overall framework is novel, but some design choices made in the framework are reminiscent of prior works, such as Dreamer [1], MTM [2], TT [3], UniMASK [4], and MaskDP [5]. I'm aware the authors have properly cited and briefly discussed these works, but it would be great if the authors could discuss the discrepancy between M^3PC and these works in detail in the related work or elsewhere, which might help highlight the unique contribution of M^3PC. For example, how is the sampling process of M^3PC different from that of TT, how is [PI] mask different from the goal-reaching mask used by MaskDP, and how are the applied return heuristics related to the ones used by prior works?
3. Line 294: it seems [PI] mask is illustrated in Figure 3 (b) instead of Figure 2.

[1] Hafner et al. Dream to Control: Learning Behaviors by Latent Imagination. ICLR 2020.

[2] Wu et al. Masked Trajectory Models for Prediction, Representation, and Control. ICML 2023.

[3] Janner et al. Offline Reinforcement Learning as One Big Sequence Modeling Problem. NeurIPS 2021.

[4] Carroll et al. Uni[mask]: Unified inference in sequential decision problems.NeurIPS 2022.

[5] Liu et al. Masked autoencoding for scalable and generalizable decision making. NeurIPS 2022.

---

> ### Author Response · Authors · 2024-11-25
> **Friendly Reminder: Discussion Period Closing Soon**
>
> Dear Reviewer cxyR,
>
> This is a kind reminder that the discussion period is set to close on **November 26, which is now less than two days away**. We hope that our responses and clarifications have fully addressed your concerns.
>
> If you have any further questions or require additional explanations, we would be more than happy to provide further explanations. Your feedback is invaluable, and we truly appreciate your engagement in this process.

---

> > ### Comment · Reviewer_cxyR · 2024-11-26
> > **Rebuttal Acknowledgement**
> >
> > I would like to thank the authors for addressing my concerns and providing supplementary results. The additional results have demonstrated the cost-efficiency of M^3PC as well as its competitive performance on long-horizon tasks. I believe these results will further strengthen this work and please include them in the final version. Above all, I will increase my rating.

---

### Public Comment · ~Zhiwei_Jia1 · 2024-11-18
**An interesting approach combining MPC and masked trajectory models**

Hi authors,

Thanks for the great work! Please consider citing [1] as a relevant work that equips masked trajectory models with the capability of dynamic test-time replanning (although not in the strict sense of MPC).

[1] Chain-of-thought predictive control, ICML 2024

---

> ### Author Response · Authors · 2024-11-22
>
> Thank you for your comment. We will carefully review [1] and consider citing it in the related work section if our paper is accepted. We appreciate your valuable suggestion!
>
> [1] Chain-of-thought predictive control, ICML 2024

---

### Meta-Review · Area_Chair_zo9M · 2024-12-19

**Metareview:**

This paper proposes to use model predictive control (MPC) to improve inference when using transformers trained for offline reinforcement learning tasks. The proposed approach is simple and effective, as demonstrated by the authors' evaluation. The overall approach is technically solid and the results were promising, and I believe it merits publication.

**Additional Comments On Reviewer Discussion:**

The reviewers had some concerns about novelty, missing baselines, etc.; most issues were addressed during the rebuttal period. Some reviewers pointed out that there are closely related approaches that have been previously proposed, but they agreed that the framework is technically solid and the results were promising, and unanimously recommended acceptance.

---

### Decision · Program_Chairs · 2025-01-22

Accept (Poster)